# Revisiting Glorot Initialization
# for Long-Range Linear Recurrences

**Noga Bar**[*†]    **Mariia Seleznova**[*‡]    **Yotam Alexander**[†]    **Gitta Kutyniok**[‡§¶♯]    **Raja Giryes**[†]

[*]Equal contribution    [†]Tel Aviv University    [‡]Ludwig-Maximilians-Universität München
[§]University of Tromso    [¶]DLR-German Aerospace Center    [♯]Munich Center for Machine Learning

## Abstract

Proper initialization is critical for Recurrent Neural Networks (RNNs), particularly in long-range reasoning tasks, where repeated application of the same weight matrix can cause vanishing or exploding signals. A common baseline for linear recurrences is Glorot initialization, designed to ensure stable signal propagation—but derived under the infinite-width, fixed-length regime—an unrealistic setting for RNNs processing long sequences. In this work, we show that Glorot initialization is in fact unstable: small positive deviations in the spectral radius are amplified through time and cause the hidden state to explode. Our theoretical analysis demonstrates that sequences of length $t = O(\sqrt{n})$, where $n$ is the hidden width, are sufficient to induce instability. To address this, we propose a simple, dimension-aware rescaling of Glorot that shifts the spectral radius slightly below one, preventing rapid signal explosion or decay. These results suggest that standard initialization schemes may break down in the long-sequence regime, motivating a separate line of theory for stable recurrent initialization.

## 1   Introduction

Recurrent Neural Networks (RNNs) have long been foundational models for sequential data, powering applications that range from time-series forecasting [Salinas et al., 2020, Rangapuram et al., 2018] to natural-language processing [Mikolov et al., 2010, Sundermeyer et al., 2015, 2012, Gers and Schmidhuber, 2001]. Recent advances in sequence modeling have centered on State-Space Models (SSMs) [Gu et al., 2021, Gu and Dao, 2023, De et al., 2024], and a prominent concurrent work has demonstrated that a class of RNNs known as Linear Recurrent Units (LRUs) [Orvieto et al., 2023] can match the performance of SSMs with simpler architectures. In both LRUs and SSMs, the recurrent blocks are linear, enabling efficient parallel-scan computations and thereby fast training and inference. The success of these models on long-range reasoning tasks reaffirms the relevance of RNNs in modern deep learning.

Despite this potential, RNNs remain difficult to train on long sequences due to the well-known problem of vanishing and exploding gradients [Bengio et al., 1994, Pascanu et al., 2013]. The core issue lies in the repeated application of the same weight matrix at every time step: even a slight spectral imbalance is exponentially amplified, causing hidden states—and hence gradients—to either explode or vanish. While deep feedforward networks suffer from similar pathologies, each layer in such architectures uses an independent weight matrix, making the dynamics of signal propagation easier to analyze and control.

In deep feedforward networks, instability is typically mitigated by principled initialization schemes such as Glorot [Glorot and Bengio, 2010] or He [He et al., 2015] initialization. These methods are derived under the infinite-width limit, where the width tends to infinity while the depth is kept fixed. In this regime, one can compute the exact signal propagation statistics and tune the weights

39th Conference on Neural Information Processing Systems (NeurIPS 2025).

variance so that activations and gradients remain of the same magnitude [Glorot and Bengio, 2010]. Although these initialization schemes have been immensely successful in practice, recent theoretical work reveals that their underlying infinite-width approximation becomes increasingly inaccurate as depth grows [Hanin and Nica, 2019, 2020, Li et al., 2021, Seleznova and Kutyniok, 2022]. This raises questions about the applicability of such schemes to RNNs, which can be viewed as extremely deep networks in time—typically of modest width but unbounded depth. As in the feedforward case, analyzing RNNs under the joint limit of infinite width and infinite input length poses significant theoretical challenges: standard random matrix theory techniques used in the infinite-width setting no longer apply when both dimensions grow [Tao, 2012]. In RNNs, this limitation is further exacerbated by the repeated application of the same weight matrix at each time step.

Yet, recent approaches to long-range RNNs continue to be guided by infinite-width heuristics. Notably, the LRU method [Orvieto et al., 2023], whose success stems from its initialization and parametrization, is inspired by the Glorot scheme. The goal of LRU initialization to construct a diagonal matrix whose eigenvalue spectrum matches that of a Glorot-initialized weight matrix. This implicitly assumes that replicating Glorot's spectral behavior is sufficient to ensure stable recurrent dynamics. In contrast, we show that standard Glorot initialization becomes unstable when iterated over long sequences in linear recurrent networks, thereby challenging this assumption.

Our main contributions are as follows:

- **Instability of Glorot:** Prior work has correctly noted that, in the infinite-width limit, the spectrum of large non-Hermitian Gaussian matrices (real or complex) used in Glorot initialization lies within the unit circle, in accordance with the circular law [Bai, 1997, Girko, 1985]. However, we highlight a subtle but critical point often overlooked in prior works: the spectral radius of such matrices converges to one from *above* (see Section 4). In other words, the largest eigenvalue typically has a modulus slightly greater than one. While this deviation is asymptotically small, it is sufficient to cause exponential growth in the hidden state over long sequences. This makes the standard Glorot initialization inherently unstable in long-sequence recurrent settings.

- **Rescaled Glorot initialization:** To address this instability, we propose a simple rescaling of the weight matrix by a dimension-dependent constant. This adjustment lowers the probability that the radius exceeds one by shifting it approximately one standard deviation below its original expectation (see Section 5). As a result, the rescaled initialization prevents exponential growth in hidden state norms, even over long sequences. Consistent with prior work, our initialization favors eigenvalues close to one from *below*, as slow signal decay is generally preferable to amplification for memory retention in RNNs [Orvieto et al., 2023, White et al., 2004, Ganguli et al., 2008]. We therefore argue that this rescaling provides a more principled and robust baseline for long-sequence modeling than the standard Glorot scheme.

- **Signal propagation over long sequences:** We analyze the hidden state norm in linear RNNs with complex Glorot initialization, beginning with a lower bound in the *finite-width, finite-sequence* setting under i.i.d. Gaussian inputs (Section 6.1). In the infinite-width-and-length regime, our analysis guarantees exponential growth of the hidden state with time (see Section 6.1.2). Notably, we show that a sequence length of $\Theta(\sqrt{n})$ (where $n$ is the width) suffices to induce explosion. In contrast, in the infinite-width but fixed-length regime, the norm of the hidden states grows only slowly with time (see Section 6.1.1), failing to reflect long-sequence behavior.

- **Numerical experiments:** We validate our theoretical findings empirically, evaluating both the statistical behavior of hidden states at initialization and the downstream performance of linear RNNs on real-world sequential data (see Section 7). As expected, standard Glorot initialization leads to signal explosion on long-range reasoning tasks, while our rescaled initialization remains stable and trainable across multiple tasks.

Overall, our work introduces a simple yet theoretically grounded modification to Glorot initialization, establishing it as a principled baseline for long-range reasoning tasks. In doing so, we underscore the often-overlooked importance of the theoretical setting—particularly assumptions about width, depth, and scaling—that underpin initialization schemes in recurrent architectures.

## 2   Related Work

Existing initialization theory primarily addresses feedforward networks with fixed depth. Extending these insights to recurrent architectures, especially in the long-sequence regime, remains an open problem. In this section, we survey works on signal propagation, stability in RNNs, and recent efforts to understand the double-scaling limit of neural networks.

**Initialization and signal propagation in feedforward networks.**   Glorot and He initialization schemes [Glorot and Bengio, 2010, He et al., 2015] were originally developed to prevent vanishing and exploding gradients in deep feedforward networks. Their theoretical justification typically relies on the analysis of signal propagation in the infinite-width, fixed-depth regime. This framework was formalized for fully connected networks by Schoenholz et al. [2016] and Poole et al. [2016], and later extended to other architectures [Xiao et al., 2018, Tarnowski et al., 2019, Gilboa et al., 2019]. Overall, signal propagation theory has had a significant impact on network design and initialization, contributing both theoretical insights and strong empirical performance.

**Initialization and stability in RNNs.**   Extending initialization theory to recurrent nets is challenging due to weight sharing across time and the unbounded effective depth introduced by repetitively applying the recurrent matrix. While signal propagation techniques have been adapted to vanilla RNNs and simple gated units in the infinite-width, fixed-length setting [Chen et al., 2018, Gilboa et al., 2019, Alemohammad et al., 2021], they do not fully capture long-sequence processing properties.

Recent work has begun to address infinite-length regimes in simplified settings, such as single-neuron or diagonal recurrent systems [Zucchet and Orvieto, 2024], revealing new instability mechanisms not accounted for by standard initialization theory. However, in the absence of a general framework, stability in earlier RNNs was often enforced through architectural interventions—such as LSTM [Hochreiter and Schmidhuber, 1997] and GRU [Cho et al., 2014] gating—alongside normalization layers [Ba et al., 2016, Gu et al., 2021], gradient clipping [Pascanu et al., 2013, Zhang et al., 2019], or spectral regularization during training [Jose et al., 2018, Zhang et al., 2018, Kanai et al., 2017]. While these methods rely on nonlinear operations, the linearity of SSMs and LRUs is crucial for computational efficiency because it enables parallel scan operations. Consequently, modern recurrent models typically avoid complex gating (as in LSTMs and GRUs) and normalization within the recurrent block. This linearity, which prevents the use of earlier stabilization techniques, can reintroduce the problem of exploding signals [Orvieto et al., 2023, Gu et al., 2021]. To mitigate this issue, modern SSMs typically adopt structured initializations with predefined spectra to preserve stability of the recurrent unit [Gu et al., 2020, Fu et al., 2023, Gu and Dao, 2023, Gu et al., 2021]. Some works also proposed orthogonal or unitary initializations [Biegun et al., 2024, Henaff et al., 2016, Mikolov et al., 2014, Le et al., 2015] to preserve the norm over time and enable stable propagation, but these can be difficult to maintain throughout training and may be suboptimal for memory retention [White et al., 2004, Ganguli et al., 2008].

Recent interest in linear RNNs (in particular, LRUs [Orvieto et al., 2023]) has renewed attention on initialization strategies for long-horizon regimes. Despite their effectiveness, these models are sensitive to initialization, and often adopt Glorot-style variance scaling as a baseline, inherited from feedforward networks. As we discussed, this practice lacks theoretical justification in recurrent contexts, especially under long sequences. A systematic understanding of initialization in linear recurrent models under such conditions remains an open problem.

**Double scaling limit and random matrix theory.**   The theoretical challenge at the heart of signal propagation in long-range RNNs is the *double scaling limit*, where both the network's width $n$ and the sequence length (or depth) $t$ become large. This regime is natural for RNNs and other deep architectures, but it lies outside the reach of standard random matrix theory tools. Classical methods—such as free probability, spectral theory, or genus expansions—assume independent matrices or fixed depth, and thus fail when the same matrix is applied repeatedly [Tao, 2012]. Although recent work has addressed the double-scaling limit in specific architectures [Hanin and Nica, 2020, 2019, Li et al., 2021, Roberts et al., 2022, Razin et al., 2024, Seleznova and Kutyniok, 2022], to the best of our knowledge, the recurrent setting has not been explored. Our work addresses this gap by characterizing the failure of Glorot-style initialization in linear RNNs under double scaling, and proposing a theoretically grounded correction.

# 3   Problem Setup

Before presenting our contributions, we review the theoretical background relevant to this work: the linear recurrent layer, Glorot initialization, and its spectral behavior in the infinite-width regime.

## 3.1   Linear Recurrent Layer

While classical RNNs rely on non-linear activations, recent work on Linear Recurrent Units (LRUs) has shown that linear recurrences can effectively model long-range dependencies in sequential data [Orvieto et al., 2023]. In addition to their empirical performance, linear RNNs allow for more tractable theoretical analyses, particularly with respect to the spectral properties of the weight matrix. In this work, we focus on this class of models.

Given an input sequence $(\mathbf{x}_1, \ldots, \mathbf{x}_t)$ with $\mathbf{x}_i \in \mathbb{R}^n$, the corresponding hidden states $(\mathbf{h}_1, \ldots, \mathbf{h}_t)$ are computed via a time-invariant weight matrix $\mathbf{W} \in \mathbb{R}^{n \times n}$ as:

$$\mathbf{h}_t := \mathbf{W}\mathbf{h}_{t-1} + \mathbf{x}_t = \sum_{k=0}^{t} \mathbf{W}^k \mathbf{x}_{t-k}, \tag{1}$$

where we set the initial state $\mathbf{h}_0 = 0$ for simplicity. Here, $t \in \mathbb{N}$ denotes the sequence length, which may be large in long-context applications. This formulation corresponds to a vanilla linear RNN without gating.

As evident from the power series expansion, each input $\mathbf{x}_{t-k}$ is amplified (or attenuated) by $\mathbf{W}^k$, meaning the overall dynamics are highly sensitive to the spectral radius of $\mathbf{W}$. This highlights the susceptibility of linear RNNs to the exploding or vanishing gradients problem [Bengio et al., 1994, Pascanu et al., 2013], especially as the sequence length increases. The initialization of $\mathbf{W}$ thus plays a critical role in determining the stability of the model over long horizons.

*Remark* 3.1. A more general formulation of the recurrent layer includes an input projection term: $\mathbf{h}_t = \mathbf{W}\mathbf{h}_{t-1} + \mathbf{B}\mathbf{x}_t$, where $\mathbf{B} \in \mathbb{R}^{n \times n}$ is a non-recurrent input matrix. In this work, we omit $\mathbf{B}$ without loss of generality, as it does not affect our main results. For analytical purposes, we treat $\mathbf{x}_t$ as the effective input, implicitly assuming $\mathbf{B}$ is the identity or absorbed into the distribution of $\mathbf{x}_t$.

## 3.2   Glorot Initialization

Glorot initialization was originally proposed for feedforward networks to preserve the variance of activations and gradients during forward and backward propagation [Glorot and Bengio, 2010]. It prescribes initializing dense weight matrices with i.i.d. entries drawn from a normal distribution $\mathcal{N}(0, \sigma^2)$, where $\sigma = \sqrt{2/(n_{\text{in}} + n_{\text{out}})}$, and $n_{\text{in}}, n_{\text{out}}$ denote the input and output widths of the layer. When applied to a square recurrent matrix $\mathbf{W} \in \mathbb{R}^{n \times n}$, this yields:

$$\mathbf{W}^{\text{glorot}} \sim \mathcal{N}(0, 1/n), \quad \text{i.i.d.} \tag{2}$$

In classical RNNs, the recurrent matrix $\mathbf{W}$ is real and non-Hermitian. However, some recent works—including the LRU architecture [Orvieto et al., 2023]—have employed complex-valued initialization, primarily because of its analytically convenient spectral properties. Motivated by this, we also consider the *complex Glorot initialization*:

$$\mathbf{W}^{\text{glorot}} = \frac{1}{\sqrt{2}} Z_1 + \frac{i}{\sqrt{2}} Z_2, \quad \text{where } Z_1, Z_2 \sim \mathcal{N}(0, 1/n) \text{ i.i.d.} \tag{3}$$

This complex formulation enables using tools from non-Hermitian random matrix theory and yields cleaner spectral statistics than its real-valued counterpart. Below, we review existing random matrix theory results on the eigenvalue distribution of $\mathbf{W}$ under both real and complex Glorot initializations.

## 3.3   Spectral Distribution of Large Gaussian Matrices

There is a substantial body of mathematical literature analyzing the eigenvalue distribution of Gaussian random matrices in the large-$n$ limit [Ginibre, 1965, Bai, 1997, Tao and Vu, 2008, Rider, 2003, Rider and Sinclair, 2014]. One of the most well-known results in this area is the *circular law*, first conjectured by Girko [1985] and later made rigorous by Bai [1997] and others:

**Theorem 3.2** (Circular law). *Assume a matrix $\mathbf{W} \in \mathbb{F}^{n \times n}$ with $\mathbb{F} \in \{\mathbb{R}, \mathbb{C}\}$ is sampled from a complex or real Glorot initialization. Then, the empirical distribution of the eigenvalues of $\mathbf{W}$, denoted $\mu_n(\lambda)$, converges almost surely to a uniform distribution on a unit disk in the complex plane:*

$$\mu_n(\lambda) \xrightarrow[n \to \infty]{a.s.} \mathcal{U}(\{z \in \mathbb{C} : |z| \leq 1\}). \tag{4}$$

This result implies that, in the infinite-width limit, the spectrum of $\mathbf{W}$ is contained within the unit disk, and the *spectral radius* converges to 1. Consequently, each term in the linear recurrence formula (Eq. (1)) maintains bounded magnitude in expectation for any fixed $k$, suggesting that Glorot initialization yields stable signal propagation in the infinite-width setting.

Although prior work reported the empirical need to rescale Gaussian-initialized matrices to prevent hidden state explosion [Gu et al., 2021], the LRU architecture [Orvieto et al., 2023] motivates its initialization scheme based on Glorot and the circular law. In particular, they aim to imitate the spectral behavior of dense Gaussian matrices via complex diagonalized initialization. This reflects a common assumption in the RNN literature that using Glorot is enough to ensure stability.

However, this assumption overlooks the fact that recurrent networks apply the same weight matrix repeatedly, and small deviations in spectral radius—though negligible at each time step—can quickly compound over long sequences. In this work, we show that *Glorot initialization is generally unstable when applied to linear recurrent layers*, especially under the double-scaling limit of large width and sequence length.

## 4 Spectral Radius Deviations in Glorot-Initialized Matrices

While the spectral radius of Glorot initialization converges to one in the infinite-width limit, recent results from random matrix theory provide a more refined characterization of the spectral edge. In particular, Rider and Sinclair [2014] and Rider [2003] derive the asymptotic statistics for the largest eigenvalue of Gaussian matrices in the real and complex cases, respectively.

**Theorem 4.1** (Spectral radius, Theorem 1.1 of Rider and Sinclair [2014] and Theorem 1 of Rider [2003]). *Assume that $\mathbf{W} \in \mathbb{F}^{n \times n}$ is sampled from real or complex Glorot (Eqs. (2) and (3)), and denote the spectral radius by*

$$|\lambda_{\max}(\mathbf{W})| := \max\{|\lambda| : \lambda \text{ is an eigenvalue of } \mathbf{W}\}. \tag{5}$$

*Then, we have the following convergence in distribution:*

$$\sqrt{4\rho_n n}\left(|\lambda_{\max}(\mathbf{W})| - 1 - \sqrt{\frac{\rho_n}{4n}}\right) \xrightarrow[n \to \infty]{d} G, \tag{6}$$

*where $\rho_n := \log(n/(2\pi(\log n)^2))$, $G$ is a Gumbel variable with CDF $F_G(x) = e^{-(1-\delta_\mathbb{R}/2)e^{-x}}$, and $\delta_\mathbb{R} \in \{0, 1\}$ is one for real $\mathbf{W}$ and zero otherwise.*

This result shows that the spectral radius converges to 1 from *above*, and the expected size of the largest eigenvalue of Glorot initialization for sufficiently large $n$ is given by:

$$\mathbb{E}\big[|\lambda_{\max}(\mathbf{W})|\big] \cong 1 + \sqrt{\frac{\rho_n}{4n}} + \frac{\gamma - \delta_\mathbb{R} \ln 2}{\sqrt{4\rho_n n}}, \tag{7}$$

where $\gamma$ is the Euler–Mascheroni constant. In words, the expectation of the largest eigenvalue exceeds one by $O(1/\sqrt{n})$. Additionally, Theorem 4.1 implies the following about the likelihood of stability:

**Corollary 4.2.** *For real and complex Glorot-initialized $\mathbf{W}$, the probability that the largest eigenvalue lies inside the unit disk satisfies:*

$$\mathbb{P}\Big(|\lambda_{\max}(\mathbf{W})| < 1\Big) \leq \mathbb{P}\Big(|\lambda_{\max}(\mathbf{W})| < 1 + \sqrt{\frac{\rho_n}{4n}}\Big) \xrightarrow[n \to \infty]{} \frac{1}{e^{1-\delta_\mathbb{R}/2}}. \tag{8}$$

Thus, even though the bulk spectrum lies within the unit disk, eigenvalues outside the unit circle are likely for large $n$, particularly in the complex case.

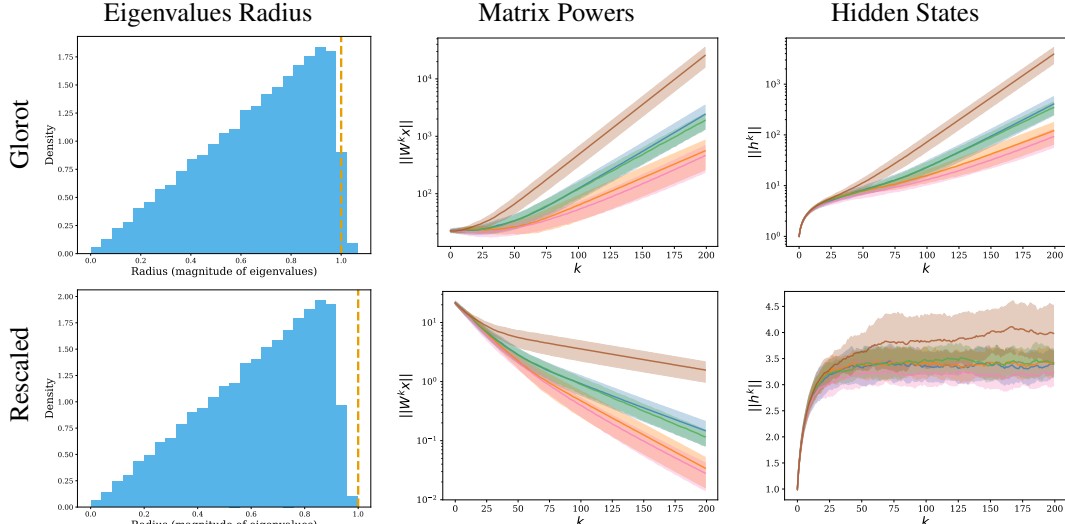

Figure 1: **Top vs bottom:** Glorot and rescaled initialization, respectively. **Left:** Empirical density of spectral radius for 100 independent samples of $\mathbf{W} \in \mathbb{R}^{500 \times 500}$, with entries drawn i.i.d. from $\mathcal{N}(0, 1/n)$. Glorot often yields eigenvalues with magnitudes exceeding one. **Middle:** Norms of $\|\mathbf{W}^k \mathbf{x}\|_2$ as a function of $k$, with each curve corresponding to a different realization of $\mathbf{W}$. Inputs $\mathbf{x}$ are sampled i.i.d. from $\mathcal{N}(0, \mathbb{I}_{500})$. We report the mean and variance over 50 random input samples. Glorot leads to exploding norms; the rescaled variant produces slowly decaying norms. **Right:** Norms of hidden states $\|\mathbf{h}_t\|_2$ in a recurrent layer with i.i.d. Gaussian inputs. The mean and variance are computed over 50 input realizations. Glorot initialization results in unstable (exploding) hidden states, while the rescaled initialization maintains stability over time.

**Connection to the double-scaling limit of neural networks.** The correction to the largest eigenvalue of order $\Theta(1/\sqrt{n})$ implies that sequence lengths on the order of $t = \Theta(\sqrt{n}^{1+\epsilon})$ are sufficient to induce exploding hidden states in linear LRUs. We make this statement precise in Section 6. This stands in sharp contrast to the circular law intuition underlying Glorot initialization, which assumes stability based on the bulk spectrum being contained in the unit disk. However, our observation fits naturally into the emerging body of theoretical work on the *double-scaling limit*, where both network width and depth grow large [Hanin and Nica, 2020, Li et al., 2021, Roberts et al., 2022]. In this regime, it has been shown that a depth scaling of $L = \Theta(n)$ suffices to cause the explosion of fourth moments in MLPs [Seleznova and Kutyniok, 2022]. Thus, our observation with respect to the spectral radius does not only reveal a flaw in the stability assumption behind Glorot initialization, but also quantifies the faster onset of instability in RNNs—arising from the repeated application of a shared weight matrix—compared to MLPs with independently sampled weights.

## 5 Rescaled Glorot Initialization for Long-Range Stability

We now introduce a corrected initialization scheme with Gaussian matrices, aimed at stabilizing signal propagation in long-sequence settings. As shown in the previous section, the spectral radius of Glorot-initialized matrices often exceeds one by a small but non-negligible amount of order $O(1/\sqrt{n})$. When the same weight matrix is applied repeatedly in linear recurrent layers, this small bias is sufficient to induce an exponential growth of hidden states. Our goal is to modify the initialization so that the spectral radius remains slightly *below* one, thus suppressing explosion while simultaneously avoiding premature vanishing.

To achieve this, we propose a variance rescaling that shifts the spectral radius to lie below one with a specified confidence level. This shift is guided by the quantiles of the Gumbel distribution, which govern the asymptotic behavior of the spectral radius under Glorot initialization (see Theorem 4.1). Let $a_p$ denote the $p$-quantile of the Gumbel distribution, that is, $\mathbb{P}(G < a_p) = p$. Then, for sufficiently

large $n$, the spectral radius of a Glorot-initialized matrix satisfies:

$$G < a_p \iff |\lambda_{\max}(\mathbf{W}^{\text{glorot}})| \lesssim 1 + \sqrt{\frac{\rho_n}{4n}} + \frac{a_p}{\sqrt{4\rho_n n}}. \tag{9}$$

Since scaling the entries of a Gaussian matrix rescales all its eigenvalues proportionally, we define our corrected initialization by reducing the variance accordingly:

$$\mathbf{W}^{\text{rescaled}} \sim \mathcal{N}\left(0, \frac{1}{n}\left(1 + \sqrt{\frac{\rho_n}{4n}} + \frac{a_p}{\sqrt{4\rho_n n}}\right)^{-2}\right) \text{ i.i.d.} \tag{10}$$

By design, this initialization has the following property for large enough $n$:

$$\mathbb{P}\left(|\lambda_{\max}(\mathbf{W}^{\text{rescaled}})| < 1\right) \cong p, \tag{11}$$

and the following holds for the expectations:

$$\mathbb{E}[|\lambda_{\max}(\mathbf{W}^{\text{rescaled}})|] = \mathbb{E}[|\lambda_{\max}(\mathbf{W}^{\text{glorot}})|]\left(1 + \sqrt{\frac{\rho_n}{4n}} + \frac{a_p}{\sqrt{4\rho_n n}}\right)^{-1} \lesssim 1. \tag{12}$$

**Choice of $a_p$ parameter.** The value of $a_p$ should be chosen so that the spectral radius is likely to remain below one, but close enough to one to avoid rapid signal decay—since vanishing dynamics limit the memory capacity. Importantly, the consequences of positive and negative deviations from the unit spectral radius are asymmetric: while a slight contraction reduces memory gradually, a slight expansion leads to rapid exponential growth due to the repeated accumulation of unstable terms in the hidden state. This asymmetry makes explosion far more damaging than mild vanishing.

In our work, we set $a_p = \gamma - \delta_{\mathbb{R}} \ln 2 + \pi/\sqrt{6}$, corresponding to one standard deviation above the Gumbel mean, and yielding $p \approx 0.86$. This choice significantly increases the likelihood that the spectral radius lies within the unit disk: from roughly $p = e^{-1/2} \approx 0.61$ in the real Glorot case and $p = e^{-1} \approx 0.37$ in the complex case, to $p \approx 0.86$ after rescaling. Notably, at the infinite-width limit, the largest eigenvalue modulus still converges to one, preserving compatibility with the circular law.

**Empirical validation.** Numerical simulations in Fig. 1 (Left) clearly show that real Glorot-initialized matrices often exhibit a spectral radius exceeding one, whereas our corrected initialization makes this much less likely. As shown in Fig. 1 (Middle), the proposed scaling also prevents the norm $\|\mathbf{W}^k\mathbf{x}\|$ from exploding for sequence lengths of order $k = O(\sqrt{n})$, in contrast to the standard Glorot initialization, where $\|\mathbf{W}^k\mathbf{x}\|$ grows rapidly. Furthermore, we compare the effect of both initializations on the evolution of the hidden states, demonstrating that our scaling mitigates instability in both $\mathbf{W}^k\mathbf{x}$ and the hidden states $\mathbf{h}_k$. A rigorous analysis covering both finite and infinite input sequences is provided in Section 6, theoretically supporting the observed numerical behavior.

## 6 Explosion of Hidden States with Long Sequences

We analyze forward signal propagation in linear recurrent layers initialized with complex Glorot. Our key result is a lower bound on the hidden state norm in the setting of *finite width and finite input length*. By examining how this bound behaves in two regimes—one where width tends to infinity with fixed sequence length, and another where both grow simultaneously—we show that hidden states remain stable in the former, but grow exponentially in the double-scaling regime.

### 6.1 Lower Bound on Hidden State Growth

Since the distribution of eigenvalues of a complex Glorot matrix is known exactly from random matrix theory and has a relatively tractable form [Tao, 2012], we can use it to lower bound the growth of the hidden state summands for finite $n$ and $k$ in the following theorem (see proof in Appendix B).

**Theorem 6.1** (Variance lower bound). *Suppose $\mathbf{W} \in \mathbb{C}^{n \times n}$ is sampled from a complex Glorot initialization (Eq. (3)), and $\mathbf{x} \sim \mathcal{N}(0, \mathbb{I}_n)$ is independent of $\mathbf{W}$. Then the following holds for any $k, n \in \mathbb{N}$:*

$$\mathbb{E}[\|\mathbf{W}^k\mathbf{x}\|_2^2] \geq \frac{1}{n^k}\frac{n}{k+1}\frac{(n+k)!}{n!}. \tag{13}$$

Suppose the inputs sequence $(\mathbf{x}_1, \ldots, \mathbf{x}_t)$ is such that each input is sampled i.i.d. from $\mathcal{N}(0, \mathbb{I}_n)$. Then the above result is directly related to the hidden state as follows:

$$\mathbb{E}[\|\mathbf{h}_t\|_2^2] = \sum_{k=0}^{t} \mathbb{E}[\mathbf{x}_{t-k}^\top (\mathbf{W}^k)^\top \mathbf{W}^k \mathbf{x}_{t-k}] = \sum_{k=0}^{t} \mathbb{E}\big[\|\mathbf{W}^k \mathbf{x}_{t-k}\|_2^2\big]. \tag{14}$$

Since the result is derived for finite width and sequence length, both the infinite-width and double-scaling regimes arise as special cases of this general setting. Finite-width results remain relatively rare in deep learning theory, as they often require tedious combinatorial arguments—such as the path counting techniques used for ReLU networks [Hanin and Nica, 2019]. In contrast, our proof avoids path counting entirely and relies instead on the distributional density of eigenvalues, enabling a concise and tractable analysis.

In the following, we examine the implications of this result for the behavior of hidden states in both the infinite-width and double-scaling regimes.

### 6.1.1 Stability with Infinite Width and Finite Length

We first consider the fixed-length and infinite-width regime, i.e. $t = O(1)$ and $n \to \infty$, which is the prevalent setting in the literature. In this regime, we get the following for a single summand of the hidden state variance, for each $k \leq t$:

$$\lim_{n\to\infty} \frac{1}{n} \mathbb{E}[\|\mathbf{W}^k \mathbf{x}\|_2^2] \geq \frac{1}{k+1} \tag{15}$$

From this, it follows that the hidden state variance lower bound is described by the harmonic series:

$$\lim_{n\to\infty} \frac{1}{n} \mathbb{E}[\|\mathbf{h}_t\|_2^2] \geq \sum_{k=0}^{t} \frac{1}{k+1} = \log(t-1) + \gamma + \frac{1}{2t} - O\Big(\frac{1}{t^2}\Big). \tag{16}$$

The hidden states exhibit logarithmic growth with respect to $t$. Although this implies that the variance of the hidden states is theoretically unbounded, in practice the growth remains mild and is unlikely to significantly disrupt stability when sequences are short. This finding aligns with existing literature on signal propagation in recurrent networks. However, as we demonstrate in the next section, this benign behavior does not extend to the more realistic scenario of long sequence lengths.

### 6.1.2 Instability with Infinite-Width-and-Length

We analyze the behavior in the infinite-length regime, which is the central focus of our work. To obtain a meaningful bound in this setting, we consider the limit as $t, n \to \infty$ with $t = \Theta(\sqrt{n})$. Under this scaling, we can use the following asymptotic approximation:

$$\frac{1}{n^k} \frac{(k+n)!}{n!} = \prod_{d=1}^{k} \frac{n+d}{n} = \prod_{d=1}^{k} \Big(1 + \frac{d}{n}\Big) \approx e^{\frac{k(k+1)}{2n}} \approx e^{k^2/2n}, \tag{17}$$

where we observed that the terms $\frac{d}{n}$ satisfy $0 < \frac{d}{n} \ll 1$ for all $d \leq k \leq t$ and vanish in the limit. This approximation can be seen as the following first-order expansion of the logarithm, and we make it more precise in Appendix B.3:

$$\log\Big(1 + \frac{d}{n}\Big) \approx \frac{d}{n} \implies \log\Big(\prod_{d=1}^{k}\Big(1 + \frac{d}{n}\Big)\Big) \approx \sum_{d=1}^{k} \frac{d}{n} = \frac{k(k+1)}{2n}. \tag{18}$$

Therefore, under standard Glorot initialization, the expected norm $\|\mathbf{W}^k \mathbf{x}\|$ grows with the power $k \leq t$ when the sequence length is sufficiently large—unlike in the infinite-width regime (Eq. (15)):

$$\lim_{\substack{t,n\to\infty \\ t/\sqrt{n}\to\alpha\in\mathbb{R}_+}} \frac{k+1}{n} \mathbb{E}[\|\mathbf{W}^k \mathbf{x}\|_2^2] \gtrsim e^{k^2/2n}. \tag{19}$$

This asymptotic growth matches empirical observations in both the real and complex settings (see Fig. 1 and Appendix A). Consequently, as we show in Appendix B.3, the hidden state norm for large $t$ and $n$ can be approximated as follows:

$$\lim_{\substack{t,n\to\infty \\ t/\sqrt{n}\to\alpha\in\mathbb{R}_+}} \frac{1}{n} \mathbb{E}[\|\mathbf{h}_t\|_2^2] \gtrsim \sum_{k=0}^{t} \frac{\exp(k^2/2n)}{k+1} \approx \frac{n}{t^2} \exp\Big(\frac{t^2}{2n}\Big). \tag{20}$$

Hence, when the sequence length $t$ is at least of order $\Theta(\sqrt{n})$, the hidden state norm begins to diverge—and this behavior becomes more pronounced under more aggressive scaling. For instance, when $t = \alpha\sqrt{n}^{1+\epsilon}$ for any $\epsilon > 0$, the lower bound diverges rapidly.

*Remark* 6.2 (Real case). While the analysis in this section focuses on the analytically tractable complex Gaussian setting, we note that real Gaussian matrices exhibit qualitatively similar behavior. This is due to the close agreement in spectral statistics between real and complex non-Hermitian Gaussian ensembles [Tao and Vu, 2008]. However, the spectral density in the real case involves more intricate expressions, making analogous derivations significantly more tedious. We provide supporting numerical simulations and an additional theoretical discussion of the real case in Appendix B.2.

To summarize, our findings underscore the need to rescale Glorot initialization in order to ensure stability during long-sequence processing, especially in the double-scaling regime. Next, we demonstrate the applicability of our findings to the training of linear recurrent networks.

## 7 Experiments

In addition to our empirical validation of signal propagation at initialization (Figure 1), we evaluate the effectiveness of our rescaled initialization on standard long-range sequence modeling benchmarks.

We conduct experiments on three classification tasks drawn from the Long Range Arena (LRA) benchmark [Tay et al., 2020]: Sequential CIFAR-10, IMDB, and ListOps. In the Sequential CIFAR-10 task, image pixels are flattened into sequences of length 3K. ListOps consists of nested arithmetic expressions over single-digit integers, with a 10-class output and maximum sequence length of 2K. IMDB is a binary sentiment classification task with input sequences of up to 8K tokens.

We compare our rescaled initialization against the standard dense Glorot initialization, and a naive non-exploding baseline of Glorot/2 (i.e., $\mathbf{W} \sim \mathcal{N}(0, 1/2n)$). Additionally, we implement diagonalized variants of these initializations, which improve computational efficiency in both runtime and memory consumption. These are more closely aligned with the LRU architecture [Orvieto et al., 2023]. To evaluate our method in the diagonal setting while preserving spectral characteristics, we sample a dense matrix and initialize the diagonal using its eigenvalues. Additionally, we include a circular-law baseline in which diagonal entries are sampled uniformly from the complex unit circle, following prior work [Orvieto et al., 2023].

Table 1: Classification accuracy for dense and diagonal linear recurrent models. Note the benefit of using our rescaling.

|  | CIFAR-10 | IMDB | ListOps |
|---|---|---|---|
| Glorot | ✗ | ✗ | ✗ |
| Glorot/2 | $76.4_{\pm 0.004}$ | $\mathbf{88.74}_{\pm 0.002}$ | $46.21_{\pm 0.045}$ |
| Rescale | $\mathbf{81.54}_{\pm 0.009}$ | $87.55_{\pm 0.002}$ | $\mathbf{47.51}_{\pm 0.011}$ |
| Diag. Glorot | ✗ | ✗ | ✗ |
| Diag. Glorot/2 | $71.6_{\pm 0.006}$ | $86.67_{\pm 0.002}$ | $49.01_{\pm 0.004}$ |
| Diag. Uniform | $\mathbf{81.47}_{\pm 0.002}$ | $86.34_{\pm 0.004}$ | $48.93_{\pm 0.004}$ |
| Diag. Rescaled | $79.53_{\pm 0.009}$ | $\mathbf{87.31}_{\pm 0.001}$ | $\mathbf{49.49}_{\pm 0.005}$ |

We release the code with the paper, and provide more details on the experimental setup in Appendix C.

**Results.** Table 1 summarizes our results. As expected, models initialized with standard Glorot failed to train due to exploding hidden states. Across the remaining baselines, our rescaled initialization achieved the highest accuracy in 4 out of 6 settings, outperforming both Glorot/2 and the unit-circle-based diagonal initialization. These results demonstrate that our simple rescaling provides a practical and theoretically grounded improvement over the standard approaches. Although these experiments are conducted on linear RNNs and do not directly compete against state-of-the-art LRUs or SSMs, they offer important initial evidence supporting theoretically-principled initialization. Additional comparisons with prior RNN stabilization techniques and an investigation of parameterization effects appear in Appendix D.

## 8 Conclusion

We revisited the widely used Glorot initialization in the context of linear RNNs and showed that, contrary to common assumptions, it leads to instability when applied to long input sequences. Our findings highlight that initialization schemes derived under the infinite-width, fixed-depth regime may not generalize to the infinite-input-length setting characteristic of recurrent architectures. This work

contributes to the growing body of theoretical research on the double-scaling limit in deep learning, and to the best of our knowledge, is among the first to explore this regime in the context of RNNs.

**Limitations and Future Work.** Our analysis focuses on the analytically tractable setting of complex Gaussian initialization in linear recurrent networks. A key limitation is that it provides only a lower bound on the hidden state norm, derived under the simplified assumption of uncorrelated Gaussian inputs. Extending this analysis to more realistic scenarios with temporally correlated inputs—and refining the bound to an exact characterization—would likely require addressing open problems in random matrix theory, particularly regarding the spectral behavior of high powers of random matrices.

Another promising direction is the development of a rigorous theoretical framework for common initialization heuristics used in practice, such as those that enforce a minimum spectral radius or constrain eigenvalues to lie predominantly along the positive real axis. A deeper understanding of these strategies could yield more stable and effective architectures for long-sequence modeling.

Finally, like many existing works, our analysis is limited to the behavior of networks at random initialization. While this provides foundational insight, a natural next step is to study the optimization dynamics of recurrent models in the infinite-input-length regime.

## Acknowledgement

N. Bar and R. Giryes thank KLA and The Center for AI & Data Science at Tel Aviv University (TAD) for supporting this research. G. Kutyniok acknowledge support by the gAIn project, which is funded by the Bavarian Ministry of Science and the Arts (StMWK Bayern) and the Saxon Ministry for Science, Culture and Tourism (SMWK Sachsen).

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

# A   Numerical Simulation for the Complex Case

Along with the experiments illustrating the behavior of the hidden state and its summands for real Glorot matrices in Section 5, we include numerical simulations for the complex case here. The results are shown in Fig. 2. As in the real case discussed in the main text, complex Glorot initialization leads to an explosion in the hidden state norm, while our proposed rescaling effectively mitigates this.

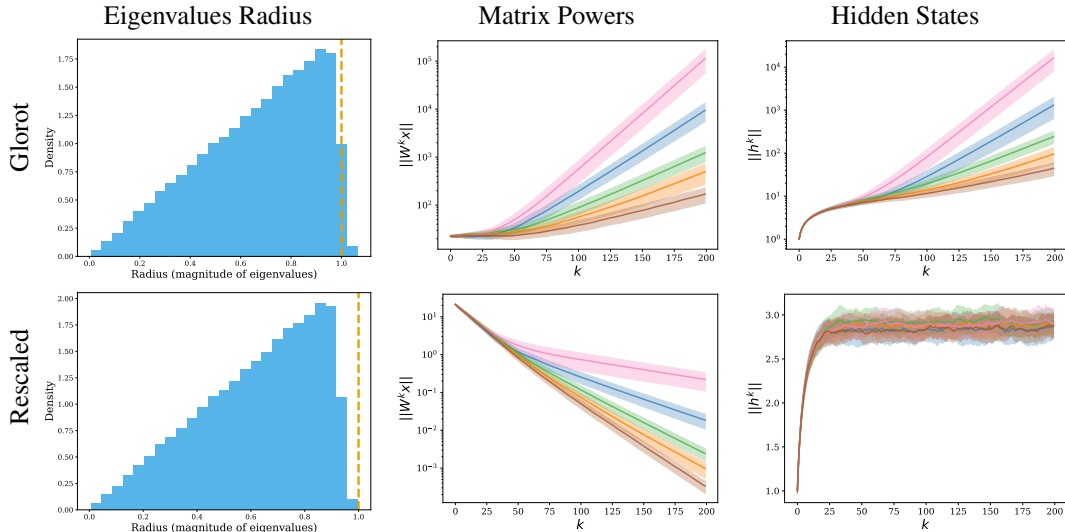

Figure 2: **Top vs. bottom:** Complex Glorot and rescaled initialization, respectively. **Left:** Empirical density of spectral radius for 100 independent samples of $\mathbf{W} \in \mathbb{C}^{500 \times 500}$, with entries drawn i.i.d. according to complex Glorot (Eq. (3)). Glorot often yields eigenvalues with magnitudes exceeding one. **Middle:** Norms of $\|\mathbf{W}^k \mathbf{x}\|_2$ as a function of $k$, with each curve corresponding to a different realization of $\mathbf{W}$. Inputs $\mathbf{x}$ are sampled i.i.d. from $\mathcal{N}(0, \mathbb{I}_{500})$. We report the mean and variance over 50 random input samples. Glorot leads to exploding norms; the rescaled variant produces slowly decaying norms. **Right:** Norms of hidden states $\|\mathbf{h}_k\|_2$ in a recurrent layer with i.i.d. Gaussian inputs. The mean and variance are computed over 50 input realizations. Glorot initialization results in unstable (exploding) hidden states, while the rescaled initialization maintains stability over time.

# B   Proofs and Additional Theory

In this section, we present formal proofs of the main theoretical results discussed in the paper. We also provide additional analysis and discussion related to real Glorot matrices, as detailed in Section B.2.

## B.1   Variance Lower Bound (Theorem 6.1)

We provide a proof for the lower bound on the growth of $\|\mathbf{W}^k \mathbf{x}\|$ in the finite-width and -length setting stated in Theorem 6.1.

**Theorem** (Variance lower bound). *Suppose* $\mathbf{W} \in \mathbb{C}^{n \times n}$ *is sampled from a complex Glorot initialization (Eq. (3)), and* $\mathbf{x} \sim \mathcal{N}(0, \mathbb{I}_n)$ *is independent of* $\mathbf{W}$. *Then the following holds for any* $k, n \in \mathbb{N}$:

$$\mathbb{E}[\|\mathbf{W}^k \mathbf{x}\|_2^2] \geq \frac{1}{n^k} \frac{n}{k+1} \frac{(n+k)!}{n!}. \tag{21}$$

*Proof.* We begin by applying a standard identity involving expectations and traces (sometimes called Hutchinson's trick):

$$\mathbb{E}[\|\mathbf{W}^k \mathbf{x}\|_2^2] = \mathbb{E}[\mathbf{x}^\top (\mathbf{W}^k)^* \mathbf{W}^k \mathbf{x}] = \mathrm{Tr}(\mathbb{E}[\mathbf{x}\mathbf{x}^\top]\mathbb{E}[(\mathbf{W}^k)^* \mathbf{W}^k]) = \mathbb{E}[\mathrm{Tr}((\mathbf{W}^k)^* \mathbf{W}^k)],$$

where we used the independence between $\mathbf{x}$ and $\mathbf{W}$, and the fact that $\mathbb{E}[\mathbf{x}\mathbf{x}^\top] = \mathbb{I}$. This reduces the problem to estimating the expected trace $\mathrm{Tr}((\mathbf{W}^k)^* \mathbf{W}^k)$.

Next, we analyze this trace via the Schur decomposition. Any complex matrix $\mathbf{W}$ admits a decomposition of the form:

$$\mathbf{W} = \mathbf{U}\mathbf{T}\mathbf{U}^{-1},$$

where $\mathbf{U}$ is unitary and $\mathbf{T}$ is upper triangular with the eigenvalues of $\mathbf{W}$ on its diagonal. This can be derived from the eigenvalue decomposition via Gram-Schmidt orthogonalization of the eigenvectors.

Raising both sides to the $k$-th power gives:

$$\mathbf{W}^k = \mathbf{U}\mathbf{T}^k\mathbf{U}^{-1},$$

and thus:

$$\mathrm{Tr}((\mathbf{W}^k)^*\mathbf{W}^k) = \mathrm{Tr}((\mathbf{T}^k)^*\mathbf{T}^k).$$

Because $\mathbf{T}$ is upper triangular, the diagonal of $\mathbf{T}^k$ consists of $\lambda_i^k$, where $\lambda_i$ are the eigenvalues of $\mathbf{W}$. Consequently, the trace splits as:

$$\mathrm{Tr}((\mathbf{W}^k)^*\mathbf{W}^k) = \sum_{i=1}^{n}|\lambda_i|^{2k} + \sum_{i>j}|(\mathbf{T}^k)_{ij}|^2 \geq \sum_{i=1}^{n}|\lambda_i|^{2k}.$$

Equality holds when $\mathbf{W}$ is normal (e.g., Hermitian), so the bound is tight.

To compute the expectation of this lower bound, we use the known eigenvalue density $\mu_n(\lambda)$ for non-Hermitian Gaussian matrices:

$$\mu_n(\lambda) = \frac{1}{\sqrt{n}\pi}e^{-n|\lambda|^2}\sum_{s=0}^{n-1}\frac{n^s|\lambda|^{2s}}{s!}.$$

This expression is due to Ginibre [1965]; see also Proposition 2.2 in Byun and Forrester [2022] for a modern treatment. Since we use Glorot initialization (i.e., standard Gaussians scaled by $1/\sqrt{n}$), we apply the change of variables $\lambda \mapsto \sqrt{n}\lambda$ to preserve the correct scaling.

Using this density, we compute the expected eigenvalue moments:

$$\mathbb{E}[|\lambda_i|^{2k}] = \frac{1}{n\pi}\sum_{s=0}^{n-1}\frac{n^s}{s!}\int_{\mathbb{C}}|\lambda|^{2(k+s)}e^{-n|\lambda|^2}d(\sqrt{n}\lambda).$$

We now evaluate the integral in polar coordinates:

$$\int_{\mathbb{C}}|\lambda|^{2(k+s)}e^{-n|\lambda|^2}d(\sqrt{n}\lambda) = \int_0^{2\pi}\int_0^{\infty}r^{2(k+s)}e^{-nr^2}\sqrt{n}r\,d(\sqrt{n}r)\,d\theta$$

$$= \frac{1}{2n^{k+s}}\int_0^{2\pi}\int_0^{\infty}(Rn)^{k+s}e^{-Rn}d(Rn)\,d\theta$$

$$= \frac{\pi}{n^{k+s}}\Gamma(k+s+1) = \frac{\pi(k+s)!}{n^{k+s}}.$$

Plugging this back in:

$$\mathbb{E}[|\lambda_i|^{2k}] = \frac{1}{n^{k+1}}\sum_{s=0}^{n-1}\frac{(k+s)!}{s!} = \frac{1}{n^k}\cdot\frac{1}{k+1}\cdot\frac{(k+n)!}{n!}.$$

Finally, summing over all $i$ and plugging into the lower bound, we obtain:

$$\mathbb{E}[\mathrm{Tr}((\mathbf{W}^k)^*\mathbf{W}^k)] \geq \frac{1}{n^k}\cdot\frac{n}{k+1}\cdot\frac{(n+k)!}{n!},$$

which completes the proof. $\qquad\square$

## B.2 Real Glorot Initialization

While our discussion of the spectral radius in Section 4 and the rescaling strategy in Section 5 covered both real and complex Gaussian initialization, the results on hidden state growth in Section 6 were presented only for the complex Glorot case. This restriction is due to the greater analytical tractability of the spectrum in the complex Gaussian setting.

Although complex-valued initialization is increasingly common in RNNs (e.g., Orvieto et al. [2023]), most classical results for feedforward networks focus on real Glorot initialization (Eq. (2)). In this section, we examine how our signal propagation results extend to the real-valued case and outline the specific analytical challenges it presents.

The main difficulty in analyzing the eigenvalue moduli for real Gaussian matrices is the lack of perfect spectral symmetry with respect to the real and imaginary axes. Unlike in the complex case, the eigenvalue distribution for real matrices is slightly biased toward the real line, and the empirical densities of real and non-real eigenvalues require separate treatment [Edelman et al., 1994]. We formalize this distinction in the following two propositions, which give the density expressions for each spectral component.

**Proposition B.1** (Real-eigenvalue density [Edelman et al., 1994, Cor. 4.3]). *Let $\mathbf{W} \in \mathbb{R}^{n \times n}$ be sampled from real Glorot initialization. Then the density of a real eigenvalue $\lambda \in \mathbb{R}$ of $\mathbf{W}$ is*

$$\mu_n^{\text{real}}(\lambda) = \frac{\sqrt{n}}{C_n^{\text{real}}} \frac{1}{\sqrt{2\pi}} \left( \frac{\Gamma(n-1, n\lambda^2)}{\Gamma(n-1)} + \frac{(\sqrt{n}\lambda)^{n-1} e^{\frac{-n\lambda^2}{2}}}{\Gamma(\frac{n}{2}) 2^{\frac{n}{2}}} \frac{\gamma(\frac{(n-1)}{2}, \frac{n\lambda^2}{2})}{\Gamma(\frac{n-1}{2})} \right),$$

*where $\Gamma(s, z) = \int_z^\infty t^{s-1} e^{-t} \, \mathrm{d}t$ is the upper-incomplete gamma function, $\gamma(s, x) = \int_0^x t^{s-1} e^{-t} \mathrm{d}t$ is the lower incomplete gamma function, and we use the shorthand $\Gamma(s) := \Gamma(s, 0)$, and $C_n^{\text{real}} = O(\sqrt{n})$ is a normalizing constant equal to the expected number of real eigenvalues.*

**Proposition B.2** (Complex-eigenvalue density [Edelman, 1997, Thm. 6.2]). *Let $\mathbf{W} \in \mathbb{R}^{n \times n}$ be sampled from real Glorot initialization. The density of a complex eigenvalue $\lambda = x + iy$ of $\mathbf{W}$ in the upper half-plane $(y > 0)$ is*

$$\mu_n^{\text{complex}}(x, y) = \frac{2n}{C_n^{\text{complex}}} \sqrt{\frac{2}{\pi}} y \, \exp\left[ n \left( y^2 - x^2 \right) \right] \, \text{erfc}\left( \sqrt{2n} \, y \right) \sum_{k=0}^{n-2} \frac{\left( n(x^2 + y^2) \right)^k}{k!}, \quad y > 0,$$

*where $\text{erfc}(x) := \frac{2}{\sqrt{\pi}} \int_x^\infty e^{-t^2} \, \mathrm{d}t$ is the complementary error function, and $C_n^{\text{complex}} = O(n)$ is a normalizing constant equal to the expected number of complex eigenvalues.*

Note that the above proposition considers only eigenvalues in the upper half-plane, as the complex eigenvalues of real Gaussian matrices occur in conjugate pairs. Consequently, the spectral distribution in the lower half-plane mirrors that of the upper half.

As a result, computing the expectation of the $2k$-th moment of the spectrum—analogous to the approach used in the proof of Theorem 6.1.1 for the complex Gaussian case—reduces to the following expression in the real case:

**Proposition B.3** (Expected $k$-th absolute moment). *For any integer $k \geq 0$, the expectation of the $2k$-th absolute moment of the spectrum of $\mathbf{W}$ can be computed as follows:*

$$\mathbb{E}\left[ \sum_{i=1}^n |\lambda_i|^{2k} \right] = \frac{C_n^{\text{real}}}{n} \underbrace{\int |x|^{2k} \, \mu_n^{\text{real}}(x) \, \mathrm{d}x}_{\text{real eigenvalues}} + \frac{C_n^{\text{complex}}}{n} \underbrace{\iint_{y>0} (x^2 + y^2)^k \, \mu_n^{\text{complex}}(x, y) \, \mathrm{d}x \, \mathrm{d}y}_{\text{complex conjugate pairs}}.$$

Unlike the complex Glorot case, there is no known closed-form expression for the corresponding integral in the real setting. Nevertheless, numerical results indicate that the behavior is remarkably similar across both cases. As shown in Fig. 3, the expected hidden state growth under real and complex initialization closely aligns. While the mean spectral radius for real matrices is slightly lower than that of complex matrices, the real case exhibits a noticeably heavier tail in its distribution.

This suggests that although real Glorot initialization is, on average, less prone to instability (as discussed in Section 4), it can lead to more rapid exponential growth when the spectral radius does exceed one.

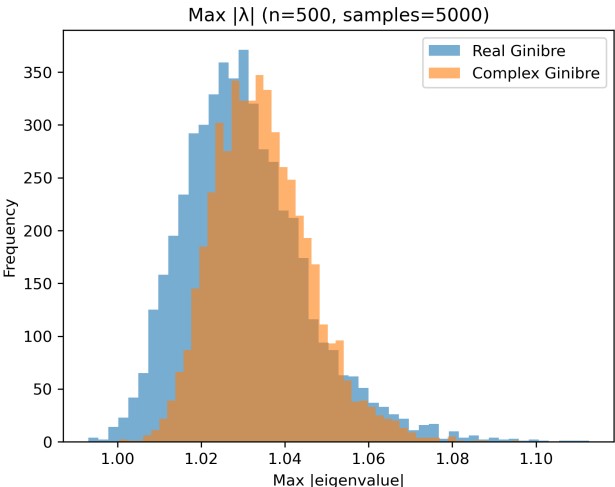

Figure 3: We show histograms of the maximal eigenvalue sizes for both the real and the complex Glorot ensembles (i.e Glorot initializations) . One can see that the behavior is similar overall, and furthermore that the upper bound in Eq. (9) is indeed satisfied with high probability (90.2 percent in the real case, and 99.2 in the complex case) **NB:** I'm not sure if the reference is correct. maybe $1 - p$ is needed . There are however two notable differences: the mean of the real Glorot matrices is slightly smaller than in the complex case, but nevertheless the right tail of the distribution is longer. While the typical size of the largest eigenvalue is similar in both cases, this latter feature results in the expected $k$-th absolute moment being larger in the real case for large $k$.

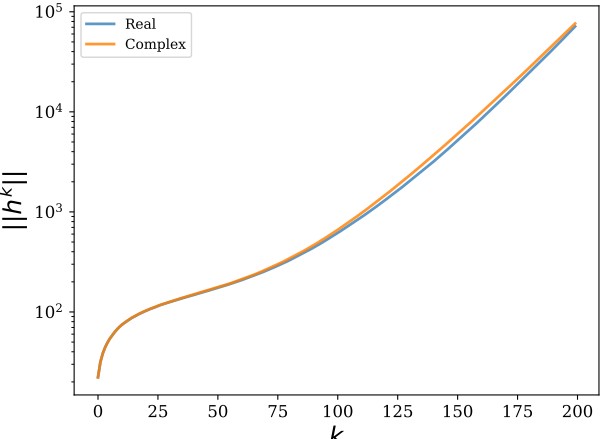

Figure 4: Norms of hidden states $\|\mathbf{h}_k\|_2$ in a recurrent layer with i.i.d. Gaussian inputs. The mean is computed over 100 realizations of $\mathbf{W}$ and 5 independent inputs. Both real and complex Glorot initializations lead to similar unstable (exploding) hidden states.

## B.3    Infinite-Width-and-Length Limit

In this section, we formally justify the asymptotic expression for the lower bound of $\|\mathbf{W}^k\mathbf{x}\|$ and $\|\mathbf{h}^t\|^2$ in the infinite-width-and-length regime. These result were introduced informally in Section 6.1.2.

We first show that the bound on $\mathbb{E}[\|\mathbf{W}^k\mathbf{x}\|^2]$ in Eq.(19) indeed holds in the double-scaling limit by the following lemma:

**Lemma B.4.** *Suppose $n, k \to \infty$ with $k = \alpha\sqrt{n}$ for some fixed $\alpha \in \mathbb{R}$. Then we have:*

$$\frac{1}{n^k} \cdot \frac{(k+n)!}{n!} \to e^{\alpha^2/2}. \tag{22}$$

This type of asymptotic expressions is well-known in mathematical analysis and arises frequently in the study of factorial approximations and exponential limits.

*Proof.* Let $k = \alpha\sqrt{n}$ for some fixed $\alpha > 0$. We begin by rewriting the expression:

$$\frac{1}{n^k} \cdot \frac{(n+k)!}{n!} = \prod_{d=1}^{k} \left(1 + \frac{d}{n}\right).$$

Taking logarithms and using the Taylor expansion $\log(1+x) = x - \frac{x^2}{2} + \cdots$, we write:

$$\sum_{d=1}^{k} \log\left(1 + \frac{d}{n}\right) = \sum_{d=1}^{k} \left(\frac{d}{n} + O\left(\frac{d^2}{n^2}\right)\right).$$

Since $d \leq \alpha\sqrt{n}$, we have $d^2/n^2 \leq \alpha^2/n$, so:

$$\sum_{d=1}^{k} \log\left(1 + \frac{d}{n}\right) = \frac{1}{n}\sum_{d=1}^{k} d + O\left(\frac{k}{n}\right).$$

Using $\sum_{d=1}^{k} d = \frac{k(k+1)}{2}$, we obtain:

$$\sum_{d=1}^{k} \log\left(1 + \frac{d}{n}\right) = \frac{\alpha^2 n + \alpha\sqrt{n}}{2n} + O\left(\frac{1}{\sqrt{n}}\right) = \frac{\alpha^2}{2} + O\left(\frac{1}{\sqrt{n}}\right).$$

Exponentiating both sides yields:

$$\frac{1}{n^k} \cdot \frac{(n+k)!}{n!} = \exp\left(\frac{\alpha^2}{2} + O\left(\frac{1}{\sqrt{n}}\right)\right) = \exp\left(\frac{\alpha^2}{2}\right)\left(1 + O\left(\frac{1}{\sqrt{n}}\right)\right),$$

which completes the proof. $\qquad\square$

We next formally justify the exponential growth of the hidden state variance $\mathbb{E}[\|\mathbf{h}^t\|^2]$ presented in Eq. (20):

**Lemma B.5.** *Fix $0 < \epsilon < \alpha$ and set $t = \alpha\sqrt{n}$. Define*

$$S_n(\epsilon, \alpha) := \sum_{k=\lfloor \epsilon\sqrt{n}\rfloor}^{\lfloor \alpha\sqrt{n}\rfloor} \frac{\exp(k^2/2n)}{k}. \tag{23}$$

*Then*

$$S_n(\epsilon, \alpha) = \int_{\epsilon}^{\alpha} \frac{e^{x^2/2}}{x}\, dx + O_{\epsilon,\alpha}\left(n^{-1/2}\right), \tag{24}$$

*and the integral admits the asymptotic estimate*

$$\int_{\epsilon}^{\alpha} \frac{e^{x^2/2}}{x}\, dx = \frac{1}{2}\left(\mathrm{Ei}(\tfrac{\alpha^2}{2}) - \mathrm{Ei}(\tfrac{\epsilon^2}{2})\right) = \frac{e^{\alpha^2/2}}{\alpha^2}\left(1 + O(\alpha^{-2})\right) + O_{\epsilon}(1), \tag{25}$$

*where $\mathrm{Ei}$ denotes the exponential integral. In particular,*

$$S_n(\epsilon, \alpha) = \frac{n}{t^2}\exp\left(\frac{t^2}{2n}\right)\left(1 + O(\tfrac{n}{t^2})\right) + O_{\epsilon}(1) + O_{\epsilon,\alpha}\left(n^{-1/2}\right), \qquad t = \alpha\sqrt{n}. \tag{26}$$

*Proof.* Let $\Delta_n := n^{-1/2}$ and $f(x) := e^{x^2/2}/x$, which is $C^1$ on $[\epsilon, \alpha]$. Writing $x_k := k/\sqrt{n}$,

$$S_n(\epsilon, \alpha) \;=\; \sum_{k=\lfloor \epsilon\sqrt{n} \rfloor}^{\lfloor \alpha\sqrt{n} \rfloor} f(x_k)\,\Delta_n, \tag{27}$$

i.e., a left Riemann sum for $\int_\epsilon^\alpha f(x)\,dx$ with mesh $\Delta_n$. By the mean value theorem on each subinterval together with endpoint rounding,

$$\left| S_n(\epsilon, \alpha) \;-\; \int_\epsilon^\alpha f(x)\,dx \right| \;\le\; \Big( (\alpha - \epsilon) \sup_{x \in [\epsilon, \alpha]} |f'(x)| + 2 \sup_{x \in [\epsilon, \alpha]} |f(x)| \Big) \Delta_n \;=\; O_{\epsilon, \alpha}(n^{-1/2}), \tag{28}$$

which proves (24).

For (25), substitute $y := x^2/2$ (so $dy = x\,dx$) to obtain

$$\int_\epsilon^\alpha \frac{e^{x^2/2}}{x}\,dx \;=\; \frac{1}{2} \int_{\epsilon^2/2}^{\alpha^2/2} \frac{e^y}{y}\,dy \;=\; \frac{1}{2}\Big( \mathrm{Ei}(\tfrac{\alpha^2}{2}) - \mathrm{Ei}(\tfrac{\epsilon^2}{2}) \Big), \tag{29}$$

where $\mathrm{Ei}(z) := \mathrm{PV}\int_{-\infty}^z e^t/t\,dt$. Using the standard asymptotic expansion $\mathrm{Ei}(z) \sim \frac{e^z}{z}\big(1 + \frac{1}{z} + O(z^{-2})\big)$ as $z \to \infty$ yields

$$\frac{1}{2}\,\mathrm{Ei}\Big(\frac{\alpha^2}{2}\Big) \;=\; \frac{e^{\alpha^2/2}}{\alpha^2}\Big(1 + O(\alpha^{-2})\Big), \qquad \frac{1}{2}\,\mathrm{Ei}\Big(\frac{\epsilon^2}{2}\Big) \;=\; O_\epsilon(1), \tag{30}$$

which gives (25). Replacing $\alpha^2$ by $t^2/n$ in the leading term gives (26). $\qquad\square$

## C  Implementation Details

Our implementation is provided anonymously at the following link, and is based on the publicly available minimal-LRU codebase originally developed by Zucchet et al. [2023]. All experiments are conducted using the Adam optimizer, with weight decay applied only to non-recurrent parameters. We employ a cosine annealing learning rate schedule, starting from a base learning rate of $10^{-3}$, with warm-up steps specified in Table 2. All models are trained with six recurrent layers. Model dimensions and additional hyperparameters are detailed in Table 2.

Table 2: Hyperparameters used for each dataset.

| Hyperparameter | CIFAR-10 | IMDB | ListOps |
|---|---|---|---|
| Warmup End | 18 | 7 | 5 |
| $D$ | 512 | 192 | 256 |
| $H$ | 384 | 256 | 192 |
| Batch Size | 50 | 32 | 32 |
| Epochs | 180 | 65 | 50 |
| LR Factor | 0.025 | 0.025 | 0.05 |
| Dropout Probability | 0.05 | 0.05 | 0 |

**Compute resources and training time.**  All experiments were conducted on a single NVIDIA GeForce RTX 2080 GPU, with peak memory usage reaching up to 9 GB. Training the full dense RNN on CIFAR-10 required approximately 26 hours, while training times for other datasets were notably shorter. The diagonal variant was more computationally efficient and trained faster; for instance, training on CIFAR-10 completed in approximately 21 hours. All our experiments are limited to 1K GPU hours on a single device.

## D  Additional Experiments

We complement the main results by comparing our approach to two stabilizing strategies for recurrent models: (i) *Layer Normalization* [Ba et al., 2016] and *LRU parametrization* [Orvieto et al., 2023].

**Layer Normalization (LN) (Table 3).** LN is a standard normalization technique in classical RNNs that stabilizes training by normalizing hidden activations within each time step [Ba et al., 2016]. In our experiments setup in Table 3, we insert LN after each recurrent matrix multiplication in otherwise linear RNN block. While LN can mitigate optimization instabilities, it breaks strict linearity in the recurrence, undermining the scan-based parallelism that makes linear models efficient.

We observe that LN enables Glorot to train but at a notable computational overhead; moreover, the final performance does not surpass our rescaled initialization without LN. For example, in the dense setting our rescaled model attains the best accuracy ($87.55\%$) while training $\approx 29\%$ faster per epoch (09:08 vs. 12:51). In the diagonal setting, LN brings Glorot to $80.64\%$, still well below our rescaled model without LN ($87.31\%$) and with additional runtime cost. Overall, these experiments suggest that while LN can mitigate exploding states for otherwise unstable inits, it sacrifices efficiency; our simple, theory-guided rescaling attains higher accuracy without normalization and preserves the linear-time scan advantages.

Table 3: Accuracy and time per epoch with IMDB dataset.

|                   | Accuracy | Training Time [min] |
|-------------------|----------|---------------------|
| Glorot Dense LN   | 86.82    | 12:51               |
| Rescaled Dense LN | 86.7     | 12:51               |
| Glorot Dense      | -        | -                   |
| Rescaled Dense    | **87.55** | **09:08**          |
| Glorot Diag LN    | 80.64    | 3:25                |
| Rescaled Diag LN  | 87.28    | 3:25                |
| Glorot Diag       | -        | -                   |
| Rescaled Diag     | **87.31** | **2:58**           |

**LRU parametrization (Table 4).** We evaluate our scheme on the complete LRU architecture [Orvieto et al., 2023], which differs from a standard linear RNN in both initialization and parametrization. Holding the LRU parametrization fixed but replacing its dedicated initialization with our rescaled scheme yields a modest drop in accuracy relative to the original LRU setup. This is expected, as the LRU parametrization and initialization were designed to be used in tandem. Nevertheless, our scheme remains competitive, highlighting that principled rescaling can transfer across parametrizations and providing a strong baseline for future co-designed variants.

Table 4: Accuracy results with LRU parametrization with IMDB dataset.

| LRU           | **87.32** |
|---------------|-----------|
| Diag. Uniform | 85.84     |
| Diag. Rescaled | 85.67    |

