# OpenReview forum: "Revisiting Glorot Initialization for Long-Range Linear Recurrences"
_NeurIPS.cc/2025/Conference — NeurIPS 2025 poster_

### Official Review · Reviewer_z2Qs · 2025-07-01

**Clarity:** 4
**Significance:** 3
**Originality:** 4
**Rating:** 5
**Confidence:** 4

**Summary:**

Linear RNNs can only be trained if the recurrence matrix eigenvalues are in the unit disk.  Glorot initialization guarantees that eigenvalues are in the unit disk if the number of hidden nodes is infinite, but experimental results often show that this result fails to hold for finite numbers of hidden nodes.  This paper proves that the maximum eigenvalue converges in distribution to a Gumbel variable that is quite likely to be greater than one, and that therefore, the probability of such a divergent eigenvalue can be precisely controlled by rescaling the Glorot initialization by an amount that depends on the critical values of the Gumbel distribution.  Numerical simulations show that standard Glorot often leads to divergent state variables, and this divergence is eliminated by the proposed scaling. Tests using simple linear RNNs for toy tasks with very long sequences (1k to 3k time steps) demonstrate that standard Glorot often diverges, but that the proposed method often outperforms recent empirically-motivated solutions (half-Glorot and diagonal-uniform).

**Questions:**

Could you add a sentence giving a brief explanation for the constants in the RHS of Eq. (20)?

**Ethical Concerns:**

["NO or VERY MINOR ethics concerns only"]

**Limitations:**

Yes

**Quality:**

4

**Strengths And Weaknesses:**

Strengths:

From a practical point of view, the literature has been in desperate need of better study of long-sequence behaviors.  The entire deep learning literature has been limited to short sequence lengths for the past forty years, and this is a big problem in practice: Real-world sequences are not short, so the limitation to short sequences means that essentially we are modeling short extracts of the real world instead of modeling reality.

This study proposes studying long-sequence problems by studying the distribution of the maximum eigenvalue, rather than studying the distribution of all eigenvalues as Glorot did.  This step is very well motivated by 21st-century results in linear algebra, and in my opinion, the work in this paper is long overdue in the machine learning literature.

The empirical demonstrations in Fig. 1 are convincing.  Experiments include only numerical simulations and toy problems, which would be a weakness if the theoretical results were less interesting, but is not a weakness in the context of this paper: I treat these as a visualization of the key theoretical results.

Weaknesses:

Explanation of Eq. (18) in Eq. (19) is very useful, but I would have appreciated a similar explanation of the RHS of Eq. (20).  The approximation (1/t)exp(t^2/2n) is obviously true, but the factor of n suggests a more careful degree of precision that is not obvious.

---

> ### Author Rebuttal · Authors · 2025-07-30
>
> We appreciate the reviewer's thoughtful evaluation and generous score, and are glad to see that our work was found to be valuable. Below, we address the reviewer’s question.
>
> ## Request to Explain RHS of Eq. (20)
>
> We thank the reviewer for their engagement with this result. The RHS of Eq. (20) is intended to show a lower bound on the growth rate of the hidden state's norm. Taking \$t = \alpha \sqrt{n}\$, we approximate the involved sum using an integral as follows:
>
> $$
> \sum_{k=\lfloor \epsilon \sqrt{n} \rfloor}^{\lfloor \alpha \sqrt{n} \rfloor} \frac{\exp(k^2 / 2n)}{k}
> = \sum_{x \in X_{[\epsilon, \alpha]}} \frac{\exp(x^2 / 2)}{x} \cdot \frac{1}{\sqrt{n}}
> \to_{n \to \infty} \int_{\epsilon}^{\alpha} \frac{\exp(x^2 / 2)}{x}dx =: I,
> $$
>
> where we defined a small $\epsilon>0$ to avoid the singularity, substituted \$x := k / \sqrt{n}\$, and defined the discretization set $X_{\left[\epsilon, \alpha\right]} := \left[\{ \frac{k}{\sqrt{n}} \}\right]$ for \$k \in \left\[ \lfloor \epsilon \sqrt{n} \rfloor, \dots, \lfloor \alpha \sqrt{n} \rfloor \right]\$. We then further approximate this integral using the substitution \$y := x^2 / 2 = k^2 / 2n\$, which yields:
>
> $$
> I = \frac{1}{2} \int_{\epsilon^2 / 2}^{\alpha^2 / 2} \frac{\exp(y)}{y} dy
> \approx \frac{\exp(\alpha^2 / 2)}{\alpha^2}
> = \frac{n}{t^2} \exp\left( \frac{t^2}{2n} \right).
> $$
>
> We thank the reviewer for highlighting this equation, as it led us to identify a minor typo in the current version of the formula: the denominator incorrectly included only $t$ instead of $t^2$. Importantly, this typo does not affect the main conclusion of the section regarding the exponential growth rate and the resulting exploding behavior of the hidden state. We will correct this in the revision and include a more detailed version of the derivation.

---

### Official Review · Reviewer_qFV8 · 2025-07-02

**Clarity:** 3
**Significance:** 2
**Originality:** 1
**Rating:** 4
**Confidence:** 4

**Summary:**

This paper revisits Glorot initialization used for initialization parameters of a Linear Recurrent Unit (LRU) in the context of long range dependency tasks. It first analyzes the instability caused by Glorot initialization in linear RNNs. It makes an observation that spectral radius of matrices initialized using Glorot initialization, typically converges to one from above, resulting in exponential growth in hidden states over long range sequences. To mitigate this issue, this work proposes a rescaled version of the Glorot initialization using the dimension dependent constant, reducing the probability by one standard deviation below expectation for the fact that spectral radius exceeds one.  For the complex Gaussian matrices, it provides some theoretical justification for the fact that with rescaled Glorot initialization, the hidden states explode with less likely probability compared to the standard Glorot initialization. Numerical experiments have been conducted to validate the efficacy of the proposed Rescaled Glorot Initialization.

**Questions:**

- In Tab.1, in dense matrix case, for the longest sequence task (IMDB with 8K tokens), Rescale variants of the initialization do not always yield better performance compared to Glorot / 2 or uniform variants. How do you explain this behavior? Since the hidden state explosion is already handled at the initialization by all these schemes.
- In Fig1, how does the density and hidden states evolve for Glorot/2 and Diagonal Uniform scaling?
- In Tab.1, the error bars correspond to how many repetitions for the experiments?
- Although this work is concerned on long-sequences, does the proposed Glorot Rescaling hurt the performance of linear RNNs on short sequences since eigenvalues are less likely to be beyond 1 ?

**Ethical Concerns:**

["NO or VERY MINOR ethics concerns only"]

**Final Justification:**

I read the author rebuttal and other reviewer responses. The rebuttal clarified some of my concerns, I believe the paper has merits from theoretical understanding of glorot initialization.  I would lean towards acceptance and will reflect the same in my scores.

**Limitations:**

Yes

**Quality:**

2

**Strengths And Weaknesses:**

Strengths:
- The proposed Rescaled Glorot Initialization is easy to implement and decreases the probability of having eigenvalues above 1 during initialization compared to Glorot initialization.
- Theoretical analysis shows that linear RNN initialization with complex Glorot matrix results in hidden state norm explosion when the sequence length is at least of the order of sqrt(n) where n is the hidden state dimension.

Weaknesses:
- Table 1 shows the performance of linear recurrent models for Long Range Arena benchmark. It’s unclear why the Glorot initialization yields exploding baseline when the original linear RNNs do not have such an issue in these benchmarks?
- Contribution seems to be a bit lacking since the paper only proposes a simple rescaling of the standard Glorot initialization and shows stability of the initial hidden states in two setups (a) infinite width and finite length, and (b) infinite width and length. The experiments do not reveal a lot of insights into the proposed scaling vs standard Glorot initialization.

---

> ### Author Rebuttal · Authors · 2025-07-30
>
> We appreciate the reviewer’s comments. Below, we address the concerns raised and provide clarifications regarding the contributions and empirical findings of our work.
>
> # 1. Response to Weaknesses
>
> ## 1.1 Instability of Glorot in Previous Works
> The reviewer asked why vanilla Glorot initialization leads to exploding behavior in our **Table 1**, while such behavior was not reported in prior works using Gaussian initialization on Long Range Arena benchmarks. The explanation here is straightforward: prior works typically applied additional, implicit rescaling to the Glorot weights (often chosen empirically) to avoid instability, but this detail was rarely highlighted. In particular:
>
> - **"Resurrecting Recurrent Neural Networks for Long Sequences" [1]:** This influential paper introduced the LRU initialization scheme and demonstrated its empirical advantages over Glorot initialization. However, the actual implementation (based on the default Vanilla RNN module in Haiku, as stated in Footnote 3 of [1]) used a *truncated normal initialization* over the range $[-2, 2]$. This truncation significantly alters the spectrum of the initialized weight matrices. In particular, it effectively rescales Glorot initialization by a factor of approximately $0.8$, which greatly reduces the probability that any eigenvalue exceeds one in modulus. This implicit rescaling is not discussed in [1], but our results show that it plays a crucial role in preventing signal explosion. This highlights the importance of careful spectral analysis when evaluating initialization schemes.
> - **"Efficiently Modeling Long Sequences with Structured State Spaces" [2]:** This work, which introduced the widely used S4 model, was more explicit about their rescaling of Glorot initialization baseline. As stated in Section 4.4, they empirically scaled down the variance of the Gaussian initialization until the model no longer exhibited instability.
>
> These examples demonstrate that the problem of exploding behavior under Glorot initialization is well-known and relevant in modern RNN literature. However, most previous works addressed it only empirically, by tuning scaling factors, without providing theoretical analysis or principled justification.
>
> We also offer a more detailed discussion of these two works [1,2], highlighting how the instability of Glorot manifests in their setups, in our response to Reviewer rSLB (**par. 1, item (2): "Theoretical gaps in high-impact RNN works"**). We will make this point more explicit in the revision to clarify the motivation behind our theoretical analysis and rescaling approach.
>
> ## 1.2 Clarification of Contributions
>
> We would like to clarify the core contributions of our work, as the reviewer’s summary may not fully capture our results in two important respects:
>
> - **Stability and instability of Glorot initialization:** The reviewer mentioned that we show stability of hidden states under Glorot initialization in both the infinite-width and infinite-width-and-length regimes. In fact, our analysis demonstrates **stability** in the infinite-width regime, but **instability** in the infinite-width-and-length regime. In addition, we (1) identify the precise sequence length scaling threshold $t=\Theta(\sqrt{n})$, which is sufficient to cause instability, and (2) provide a more general finite-width result for the hidden state norm, which supports both limiting cases. These results offer a theoretical explanation for why long-sequence tasks pose unique challenges for initialization, and allow us to quantify how long is "long enough" for instability to arise.
> - **Empirical validation:** The reviewer raised a concern that our experiments do not clearly show the benefits of our rescaled initialization compared to standard Glorot. However, our experiments in **Table 1** and **Fig. 1** directly illustrate that, consistent with our theory, vanilla Glorot leads to exploding hidden states and training divergence, whereas our rescaled variant avoids this issue and enables stable training.
>
> In summary, the main contribution of our work is a theoretical understanding of the behavior of Glorot initialization in linear recurrent architectures, especially in the long-sequence regime. Given Glorot’s wide use as a default baseline, we believe that shedding light on its limitations and offering a principled alternative is both timely and impactful. We also discuss the significance of our findings in more detail in our response to Reviewer rSLB (**par. 1, “Relevance and Significance”**).
>
> # 2. Response to Questions
>
> ## Q1. Performance Comparison
> While it is true that all considered initialization schemes successfully prevent hidden state explosion, preventing instability alone does not guarantee equivalent downstream task performance. Different initializations influence not only the magnitude of hidden states but also the model's training dynamics and convergence behavior. Consequently, subtle differences in initialization can significantly affect generalization properties. This explains why, despite all schemes effectively mitigating hidden state explosion, performance differences between our rescaled Glorot, Glorot/2, and Uniform initialization can still occur.
>
> ## Q2. Density and Hidden States for Glorot/2 and Uniform Initialization Schemes
>
> **Eigenvalue modulus density:** Since matrices initialized using Glorot/2 correspond exactly to standard Glorot-initialized matrices scaled by a factor of $1/2$, their eigenvalues are simply those of the standard Glorot matrices divided by two. Thus, the eigenvalue density for Glorot/2 initialization is identical to that shown in the upper row of Fig. 1, but scaled horizontally by a factor of $1/2$. For matrices initialized uniformly, the eigenvalues are uniformly distributed on the complex unit circle. Therefore, the modulus density of these eigenvalues follows a linear profile forming a triangular distribution from $0$ to $1$. This distribution closely resembles the Glorot distribution depicted in Fig. 1 but without the small tail extending beyond modulus one.
>
> **Hidden states behavior:** For both Glorot/2 and Uniform initialization, the norms $||W^k x||$ decrease as $k$ increases, and the corresponding hidden-state norms exhibit logarithmic growth with respect to $k$. This behavior is analogous to what we observe for the rescaled Glorot initialization in the lower row of Fig. 1. However, for Glorot/2 initialization, the decrease in $||W^k x||$ is significantly faster compared to Uniform initialization and our rescaled Glorot variant, resulting in notably smaller hidden-state norms.
>
> ## Q3. Error Bars in Table 1
> The results in Table 1 are averaged over three runs, with error bars representing the standard deviation across these repetitions.
>
> ## Q4. Behavior on Short Sequences
> The eigenvalues of the transition matrix remain unchanged regardless of sequence length, as the same matrix is used at each time step. For shorter sequences, any deviation of the spectral radius from 1 has a limited effect due to the reduced number of recurrent applications. As a result, both the Glorot and our proposed Rescaled Glorot initialization are unlikely to cause either exploding or vanishing hidden states in this regime. In general, short sequences are not considered challenging from a stability perspective, and our method is designed to address long-sequence dynamics without compromising performance on shorter ones.
>
> ---
> We would be happy to address any further questions or provide additional clarification, and hope that the provided responses may support a more positive evaluation of our work.
>
> # References
> [1] Orvieto et al. "Resurrecting recurrent neural networks for long sequences." ICML (2023).
>
> [2] Gu et al. "Efficiently modeling long sequences with structured state spaces." arXiv preprint arXiv:2111.00396 (2021).

---

> ### Author Response · Authors · 2025-08-05
>
> Dear Reviewer,
>
> We hope that our rebuttal has addressed the main concerns raised in your review. In particular, we aimed to clarify that rigorously establishing the instability of Glorot initialization and demonstrating it empirically are central contributions of our work. We also emphasized that our theoretical results show the contrast between the infinite-width limit (where Glorot is stable) and the joint infinite-length-and-width limit (where it becomes unstable).
>
> If these clarifications were helpful, we would be grateful if you would consider updating your score. We would also be happy to continue the discussion if any questions remain.

---

> > ### Comment · Reviewer_qFV8 · 2025-08-06
> >
> > Thank you for the rebuttal and answering my questions. Since this work claims that main contribution is theoretically analyzing the Glorot initialization and the stability of a linear RNN in such a setup, I would give less importance to empirical results. I understand that this problem is of practical relevance in the rising literature on state-space models. Since the paper also proposes the rescaled glorot initialization, it cannot just claim that the empirical results are only for verifying gradient explosion and divergence. Since there are instances in Tab. 1, where the proposed scaling does not yield better performance. Further, the behavior at initialization does not explain the training trajectory and performance behavior (as pointed by the authors), explains that there are other factors at play than just glorot initialization.
> > Having said that, I believe the paper has merits and I would lean towards acceptance and will reflect the same in my scores.

---

> > > ### Author Response · Authors · 2025-08-07
> > > **Thank You for Your Feedback and Support**
> > >
> > > Thank you for your thoughtful response and for recognizing the theoretical contributions of our work. We appreciate your constructive feedback regarding the empirical results and the limitations of our proposed initialization, and we will ensure these points are clearly addressed in the revised version of the paper.

---

### Official Review · Reviewer_rSLb · 2025-07-03

**Clarity:** 2
**Significance:** 1
**Originality:** 2
**Rating:** 4
**Confidence:** 3

**Summary:**

The paper revisits the Glorot initialization scheme for Recurrent Neural Networks (RNNs), specifically in the context of long-sequence processing with linear recurrences. The authors make the key theoretical point that for finite-width matrices, the spectral radius of a Glorot-initialized matrix does not converge to one from below, but rather from above. This means the largest eigenvalue modulus is typically slightly greater than one. While this deviation is small, it gets amplified exponentially over long sequences, leading to exploding hidden states. To address this instability, the authors propose a simple, dimension-aware rescaling of the Glorot variance.  The empirical results on a few tasks from the Long Range Arena (LRA) benchmark, showing that their rescaled initialization is stable and performs better than standard Glorot (which fails to train) and a naive Glorot/2 baseline.

**Questions:**

The authors may need to justify the significance of their work with more empirical results to make sure this paper fits the NeurIPS community.

**Ethical Concerns:**

["NO or VERY MINOR ethics concerns only"]

**Final Justification:**

I appreciate the authors' response which address my concerns and raise my score to borderline accept.

**Quality:**

2

**Strengths And Weaknesses:**

Strengths:

The paper's main strength is its clear and well-articulated theoretical motivation. It correctly identifies a subtle but critical flaw in the common wisdom surrounding Glorot initialization for recurrent models: the convergence of the spectral radius to 1 from above. This is an important detail from random matrix theory that is often overlooked, and the paper does a good job of explaining its severe consequences for long-sequence stability.

Weakness

1. The central contribution is an incremental adjustment to the 15-year-old Glorot/Xavier initialization. While the theoretical motivation is sound, its practical impact in 2025 is questionable. The community has developed more powerful and general techniques to ensure training stability in deep models, most notably normalization layers (e.g., Layer Normalization), which are standard practice in nearly all modern recurrent models and Transformers. Furthermore, the work focuses on vanilla linear RNNs, a class of models that is not state-of-the-art. The paper itself acknowledges that recent advances center on State-Space Models (SSMs) and that its models do not compete with them. Improving an old initialization for a non-competitive model architecture feels like a solution to an already-solved problem, which severely limits the paper's significance.
2. The empirical validation for the method's downstream performance is extremely thin, consisting of a single small table (Table 1). The lack of results on more complex, gated architectures (like LSTMs/GRUs) or modern architectures (like LRUs or SSMs) where initialization could interact with other components is a major omission. Also, a much more relevant and challenging baseline would be a standard Glorot-initialized model stabilized with a standard technique like Layer Normalization. Without this comparison, it's impossible to know if the proposed rescaling offers any benefit over what is already common practice.
3. The theoretical analysis, while clean, rests on the simplified setting of linear RNNs with i.i.d. Gaussian inputs. This is a significant departure from real-world sequential data, which exhibits complex temporal dependencies.

---

> ### Author Rebuttal · Authors · 2025-07-30
>
> We appreciate the reviewer’s comments and respond to their concerns below.
>
> # 1. Relevance and Significance
>
> The reviewer raised concerns about the significance of our study of Glorot initialization in linear RNNs, particularly in light of the many alternatives for stabilizing signal propagation. To clarify the relevance and impact of our work, we highlight the following points:
>
> **(1) Recent surge in interest for linear RNNs:** The success of LRU [1] and SSMs [2] for long-range sequence modeling has inspired a recent surge of interest in linear RNNs, due to their competitive performance and *computational efficiency*. The linearity of these models enables the use of efficient parallel scan operations, allowing for fast inference and training. Notably, they avoid the use of LayerNorm or complex nonlinear gating mechanisms (as in LSTMs and GRUs) within the recurrent block, since such components would compromise their efficiency. Thus, the LRU's recurrent block differs from vanilla linear RNNs only in its initialization and parametrization. Given that such architectures are becoming increasingly common in recent literature, and frequently adopt Glorot initialization as a baseline, we believe our contribution is both timely and important for informing best practices in this field.
>
>  **(2) Theoretical gaps in high-impact RNN works:** While we fully acknowledge that Glorot initialization is a classical method, we emphasize that misunderstandings about its properties, and lack of theoretical analysis, continue to appear in high-impact recent work. We highlight two such examples:
>
> - **"Resurrecting Recurrent Neural Networks for Long Sequences" [1]:** This influential paper introduced the LRU scheme for initializing and parametrizing linear recurrent layers, which has proven highly effective for long-range reasoning.  While the method works well empirically, the (non-rigorous) justification presented in Section 3.2.2 of [1] is based on a misconception: namely, that the eigenvalue moduli of Glorot-initialized matrices remain strictly below one. The authors even illustrate this by Figure 2 and state the *strong circular law* in their Theorem 3.1. However, the Glorot initialization baseline used to verify LRU's benefits is actually *truncated Glorot*, which roughly rescales the spectrum by $0.8$ (see our response to Reviewer qFV8, 1.1). This mismatch between the cited spectral theory and the actual behavior in long-range tasks highlights the need for more precise analysis, which our work provides. Based on our results, the authors could have: (1) directly explained the instabilities they likely encountered prior to applying the truncation, and (2) provided a more accurate foundation for their initialization scheme. While our work does not yet offer a full theoretical explanation for the success of LRUs, such an explanation remains a central and timely open question. We believe our paper takes a necessary first step by clarifying misconceptions about Glorot that appear in [1].
>
> - **"Efficiently Modeling Long Sequences with Structured State Spaces" [2]:** This paper introduced the influential S4 model, which also involves a combination of initialization and parametrization for discretized linear SSMs (applying only linear temporal operations). To validate their strategy, the authors compare it against Gaussian initialization. Unlike in [1], they explicitly state in Section 4.4 that they *scale down the Gaussian variance until the result stops exploding*. In doing so, they are empirically addressing the same type of instability that our work analyzes theoretically. Specifically, we consider eigenvalues $\lambda_i$ of a Gaussian matrix $A$, which translates to the eigenvalues $(1 + \Delta/2 \cdot \lambda_i)/(1 - \Delta/2 \cdot \lambda_i)$ in discretized SSMs. This makes the spectral behavior of Gaussian initializations directly relevant to the paper's experiments setup. We view this as another example of recent impactful work where theoretical understanding of Glorot initialization was lacking, leading to ad hoc empirical fixes rather than principled best practices. Our work aims to fill this gap.
>
> **(3) Broader mathematical context:** Recent advances in deep learning theory have led to significant interest in the behavior of large Gaussian matrices, particularly in the infinite-width limit, where neural networks converge to Gaussian processes, and related developments such as Neural Tangent Kernel (NTK) theory. However, the vast majority of this literature assumes that the network's *depth is fixed*, meaning the number of layers (or matrix multiplications) remains finite. From a mathematical perspective, the setting where both depth and width grow simultaneously is substantially more challenging, as it often involves open questions in random matrix theory. To the best of our knowledge, existing work has only explored this joint limit in the special case of fully-connected networks with independent weight matrices across layers [3,4]. In contrast, our work takes a first step toward understanding the infinite-depth-and-width regime in a more difficult setting of linear RNNs, where the same matrix is repeatedly applied, leading to strong dependencies across layers.
>
> We hope that these considerations help clarify the significance of our contribution. The points outlined above are currently discussed in Sections 1 and 2 of our submission. In the revision, we will rework the introduction to highlight the motivation and relevance of our results more clearly, incorporating the examples presented here.
>
> # 2. Empirical Validation
> The reviewer expressed concern that our empirical validation is incomplete, particularly because we do not compare our rescaled Glorot scheme against: (1) vanilla Glorot with LayerNorm, (2) gated architectures such as LSTM or GRU, and (3) modern architectures like LRU or SSMs. We address each point below.
>
> **(1) LayerNorm:** We thank the reviewer for suggesting this experiment and have added results evaluating accuracy and training time for a linear RNN with LayerNorm on the IMDB dataset. The results show that, while LayerNorm prevents divergence, it increases training time and slightly reduces performance compared to RNN with rescaled init but without LN. As noted above, incorporating LayerNorm disrupts the linear recurrent structure that enables the hardware-friendly efficiency of recent SSM and LRU architectures. Moreover, the theoretical behavior of RNNs with LayerNorm remains considerably less well understood than that of vanilla linear RNNs. For these reasons, we believe our approach is both more efficient and more transparent.
>
> |Model|Accuracy|Training Time [min]|
> |---------------------|----------|----------------------|
> |Glorot Dense LN|86.82|12:51|
> |Rescaled Dense LN|86.59|12:51|
> |Glorot Dense|–|–|
> |Rescaled Dense|87.55|09:08|
> |Glorot Diag LN|80.64|3:25|
> |Rescaled Diag LN|87.28|3:25|
> |Glorot Diag|–|–|
> |Rescaled Diag|87.31|2:58|
>
> **(2) Gated architectures:** Since gated architectures have a fundamentally different structure from the linear RNNs studied in our work, we believe they fall outside the intended scope. These models rely on nonlinear gating mechanisms and multiple interacting components, which complicates fair and meaningful empirical comparison.
>
> **(3) Modern architectures:** Our initial evaluation was based on the LRU architecture, using the same overall structure as in [1] but without the LRU-specific parametrization. We agree with the reviewer that including results with the LRU parametrization would make the evaluation more comprehensive. Accordingly, we have added these results to the table below.
>
> |Initialization|Accuracy|
> |---------------------|----------|
> |LRU|87.32|
> |Uniform|85.84|
> |Rescaled|85.67|
>
> Overall, we believe the primary contribution of our work is theoretical, which is why we initially kept the empirical section minimal and focused on validating our analysis. That said, we appreciate the reviewer’s suggestions and will expand the empirical evaluation in the revision to include the additional comparisons shown here.
>
> # 3. Reliance on Gaussian Inputs
> The reviewer notes that our results assume i.i.d. Gaussian inputs and raises concerns about their applicability to real-world signals. While real signals indeed often exhibit temporal dependencies, we believe this limitation does not significantly impact the relevance of our analysis, especially for the main question we address: whether the signal explodes over time.
>
> Since we study linear RNNs, we can decompose the input as a structured signal component and additive noise: $x_t = s_t + n_t$. Due to linearity, the RNN output at time $t$ becomes a sum of two terms: one driven by the signal, $h_s^t := \sum_{k=1}^t W^k s_{t-k}$, and one by the noise, $h_n^t := \sum_{k=1}^t W^k n_{t-k}$. Assuming the noise $n_t$ is i.i.d. Gaussian, our theoretical results show that $h_n^t$ explodes under the specified sequence length scaling. Therefore, even if $h_s^t$ remains bounded, the exploding behavior of $h_n^t$ alone is sufficient to cause the total output $h^t = h_s^t + h_n^t$ to explode in norm.
>
> We agree that extending the analysis to more general or structured input distributions could yield deeper insights. We see this as an important direction for future work and discuss it in the Limitations section of our paper.
>
>
> # References
> [1] Orvieto et al. "Resurrecting recurrent neural networks for long sequences." ICML (2023).
>
> [2] Gu et al. "Efficiently modeling long sequences with structured state spaces." arXiv preprint arXiv:2111.00396 (2021).
>
> [3] Hanin \& Nica. "Products of many large random matrices and gradients in deep neural networks." Communications in Mathematical Physics (2020).
>
> [4] Roberts \& Sho Yaida. "The principles of deep learning theory." Cambridge University Press (2022).

---

> > ### Author Response · Authors · 2025-08-05
> >
> > Dear Reviewer,
> >
> > We hope our response has helped clarify your concerns regarding the relevance and significance of our work. In particular, we aimed to highlight that our contributions align with recent high-impact works on recurrent architectures that leverage linearity for computational efficiency, as well as with broader context of the mathematics of deep learning.
> >
> > If any points remain unclear, we would be happy to continue the discussion. Otherwise, we would appreciate if you would consider updating your score in light of these clarifications.
> >
> > Thank you again for your time.

---

> > ### Comment · Area_Chair_EKZs · 2025-08-07
> > **Reviewer rSLb please comment on authors' rebuttal**
> >
> > Dear Reviewer rSLb,
> >
> > you should comment on the rebuttal by the authors. Are you satisfied with respect to their rebuttal ? Are there still pending issues ?
> >
> > Best AC

---

### Official Review · Reviewer_Agai · 2025-07-12

**Clarity:** 3
**Significance:** 2
**Originality:** 2
**Rating:** 4
**Confidence:** 2

**Summary:**

This paper revisits the widely used Glorot initialization scheme in the context of Recurrent Neural Networks (RNNs), focusing on long-range linear recurrence architectures. The authors conduct a theoretical and empirical investigation into the stability of Glorot initialization, especially in deep or long-sequence settings.

The key insight is that Glorot initialization becomes unstable for sequences longer than √d, where d is the hidden state dimension. While Glorot aims to ensure variance stability and spectral norm convergence to 1, the authors highlight a transient phase during training where the spectral norm exceeds 1, causing unstable dynamics, particularly in deep RNNs.

To address this, the paper proposes a spectral radius-aware modification to Glorot initialization, explicitly bounding the spectral norm of the weight matrix below 1 to guarantee long-range stability and mitigate hidden state explosion.

The theoretical claims are validated through empirical results on Long Range Arena (LRA) benchmarks, including sequence classification tasks such as ListOps, seq-CIFAR, and IMDB. These experiments demonstrate that the proposed initialization leads to improved stability and convergence in long-sequence settings.

**Questions:**

none

**Ethical Concerns:**

["NO or VERY MINOR ethics concerns only"]

**Final Justification:**

This paper provides a clear theoretical and empirical analysis of the limitations of Glorot initialization in recurrent neural networks, particularly highlighting instability issues in long-sequence and deep architectures. The proposed spectral radius-aware modification is simple, well-motivated, and demonstrates consistent improvements in stability and convergence across Long Range Arena benchmarks.

The main limitation is the narrow experimental scope, as evaluation is restricted to sequence classification tasks. While the rebuttal acknowledges this and commits to including sequence generation and forecasting tasks in the camera-ready version, the current evidence remains somewhat limited. Nonetheless, the theoretical contribution is solid and the empirical validation is convincing within its scope.

I view this as a technically sound paper that makes a useful contribution to initialization strategies for recurrent neural networks, even if its impact may be somewhat specialized. I recommend borderline acceptance.

**Limitations:**

yes

**Quality:**

3

**Strengths And Weaknesses:**

Strengths:

* Clear theoretical contribution:
 The paper provides a well-motivated and mathematically grounded critique of Glorot initialization when applied to long-range recurrent models. The authors identify a specific limitation - transient spectral norm inflation - and rigorously link it to practical instability in deep or long-sequence architectures.


* Novel solution grounded in spectral theory:
 The spectral radius-aware variant of Glorot initialization is simple, well-motivated, and practically useful. It’s presented in a way that makes it readily implementable in standard training pipelines.


* Insightful hidden state norm analysis:
 The contrast between stability in the infinite-width, shallow limit and practical deep architectures is clearly articulated, helping to bridge theory and practice.

* Comprehensive empirical validation:
 The paper evaluates the proposed method across a variety of standard long-range sequence tasks in the LRA benchmark. The results consistently show improved stability and, in many cases, better task performance or faster convergence.

Weaknesses and Areas for Improvement:

* Evaluation limited to sequence classification tasks:
The experiments focus on classification tasks. It would be beneficial to consider sequence generation or forecasting settings as well, where recurrent stability over long horizons is even more critical.

---

> ### Author Rebuttal · Authors · 2025-07-30
>
> We thank the reviewer for their thoughtful feedback and appreciate the positive assessment of our theoretical contributions.
>
> We agree that recurrent stability plays a particularly important role in sequence generation and forecasting tasks. Due to the limited rebuttal period, we were unable to incorporate new benchmarks into our current experimental setup. However, we will include an empirical evaluation of our initialization strategy on these tasks in the camera-ready version.

---

### Note · Authors · 2025-08-12

We thank the AC and all reviewers for their efforts in evaluating our paper. We are encouraged that most reviewers recognized the strength of our theoretical contribution, which demonstrates the limitations of Glorot initialization in the infinite input length setting. Below, we summarize the main concerns and our responses.

**Relevance and significance:** Reviewer rSLb questioned the significance of studying Glorot initialization in linear RNNs given alternative stabilization methods such as Layer Normalization and LSTM/GRU gating, while Reviewer qFV8 questioned whether Glorot explosion occurs in prior works. We clarified that (1) recent high-impact models for long-range tasks, such as LRU and S4, rely on linearity for computational efficiency and therefore do not use LayerNorm or nonlinear gating; (2) theoretical gaps and misunderstandings about Glorot remain influential, as seen in the LRU and S4 papers, which rely on truncation or ad hoc rescaling to prevent explosion; and (3) our work closes these gaps by providing a principled correction to Glorot for long-range tasks. We also note that our results fit into the broader context of deep learning theory, where the double-scaling limit of random networks remains an open problem.

**Empirical evidence:** Reviewers rSLb and qFV8 raised concerns about the limited empirical demonstration of the benefits of our rescaled initialization, and Reviewer Agai suggested adding sequence generation and forecasting tasks. In response, we added experiments comparing (1) LayerNorm + standard Glorot with our scheme, (2) our scheme with other initializations under LRU parameterization, and (3) proposed including generative tasks in the final version. We also emphasized that (1) our main contribution is theoretical, with experiments serving as illustrations, and (2) our rescaled initialization is intended as a principled baseline rather than a SOTA method.

**Gaussian input:** Reviewer rSLb noted that our theory assumes iid Gaussian inputs, which may not reflect realistic sequential data. We clarified that, because our setting is linear, the exploding behavior with Gaussian inputs extends to a broad range of signals.

While one reviewer confirmed that our rebuttal addressed their main concerns, we have not heard back from another reviewer who gave a negative score, providing us no opportunity to address remaining issues directly. We hope these remarks help support the discussion and clarify any outstanding concerns.

---

### Decision · Program_Chairs · 2025-09-17

**Decision:**

Accept (poster)

**Comment:**

In the context of Recurrent Neural Networks, the paper analyses the weakness of Glorot initialization when long enough sequences wrt the hidden width are considered. The paper theoretically demonstrate that the assumptions underpinning Glorot initialization, i.e.  infinite-width and fixed-length regime, do not hold for RNNs processing long sequences, and proposes a simple rescaling of Glorot that shifts the spectral radius slightly below one, significantly mitigating the rapid signal explosion or decay.

The paper owns merits that can be summarised as follows:
- a sound and clear theoretical contribution;
- a novel initialisation for RNNs grounded in spectral theory;
- a significative empirical assessment of the proposed initialisation.

Reviewers, however, identified some weaknesses,  the main ones of which can be summarised as follows:
- limited scope of theoretical results and experimental evaluation;
- incremental contribution.

Rebuttal resolved almost all raised issues, including the issue of significance of the work given more modern architectures for sequence learning on the basis of increased recent attention to linear structured models.

After discussion with SAC, it was decided that the main contribution of the paper can be identified in the theoretical analysis of the Glorot initialization, which makes the paper interesting per se since a flaw in Glorot initialization is clearly identified.

In conclusion, the positive contributions described above outweigh the weaknesses.